# Deep learning prediction of noise-driven nonlinear instabilities in fibre optics

Yassin Boussafa [1], Lynn Sader [1], Van Thuy Hoang [1], Bruno P. Chaves [1], Alexis Bougaud [1], Marc Fabert [1], Alessandro Tonello [1], John M. Dudley [2,3], Michael Kues [4] & Benjamin Wetzel [1] ✉

Machine learning is bringing revolutionary approaches into many fields of physics. Among those, photonics enables fast and scalable information processing. Photonics platforms further possess rich nonlinear dynamics that drive fundamental interest but also prove powerful for applications in computation, imaging, frequency conversion, source development and advanced signal processing. However, incoherent processes of nonlinear optics are hardly exploited in practice as the control of noise-driven dynamics remains challenging. Here, we exploit deep learning strategies and demonstrate that coherent optical seeding can effectively shape incoherent spectral broadening. We focus on the intricate interplay between weak coherent pulses and broadband noise, competing during nonlinear fibre propagation within an amplification process known as modulation instability. We demonstrate artificial neural networks' capability to efficiently predict these complex incoherent dynamics, both numerically and experimentally. Our results show that input seed properties can be inferred from the incoherent output signal. Furthermore, our approach enables reliable prediction of output spectral fluctuations, paving the way to tailoring complex photonic signals with specific correlation features.

Machine Learning (ML) is revolutionizing science, marking a significant paradigm shift from traditional numerical optimization to generalized learning strategies. The emergence of Artificial Neural Networks (ANN) and Deep Learning (DL)[1–3], has had a significant impact across the natural sciences, in particular within fields aiming to study and forecast chaotic behaviours in complex systems[4–6]. In the realm of photonics, ML is proving especially valuable, both for understanding complex propagation dynamics and for processing extended datasets[7]. On the one side, photonic architectures provide a hardware backbone to enable physical deep learning[8–10], and efficient computation with limited power consumption[11–17]. On the other side, the domain of nonlinear and ultrafast optics is characterized by large datasets and inherent complexity, making deep learning a valuable addition for

analysis and applications[18,19]. Specific areas of demonstrated application include laser physics[20–23] and optical signal processing, in particular pulse shaping[24,25], metrology[26] and telecommunications[27–29]. In nonlinear optics, recent advances have also underlined the tremendous potential of ML frameworks for the identification of key parameters or the computation of complex propagation schemes. For example, ML can be used for the control of nonlinear spectral broadening in supercontinuum generation[30–34] and tailored frequency combs[35–38]. Deep learning can also be leveraged to forecast dynamical evolution in nonlinear pulse propagation: neural networks have been used to predict pulse reshaping in nonlinear fibre propagation for a variety of initial conditions[39–41] and to directly model pulse evolution by emulating standard computation tools with an excellent accuracy

[1]XLIM Research Institute, CNRS UMR 7252, University of Limoges, 87060 Limoges, France. [2]Université Marie et Louis Pasteur, CNRS Institut FEMTO-ST, 25030 Besançon, France. [3]Institut Universitaire de France, Paris, France. [4]Institute of Photonics and Cluster of Excellence PhoenixD, Leibniz University Hannover, 30167 Hannover, Germany. ✉e-mail: benjamin.wetzel@xlim.fr

and drastically enhanced speed[42–44], yet requiring prior training on existing dataset.

Despite these successes, however, standard DL frameworks still face practical limitations in nonlinear optics. For instance, rapid acquisition of large datasets with sufficient signal-to-noise ratio and long-term stability is challenging, and faces experimental limitations in bandwidth, dynamic range and resolution. Although such limitations can be partially mitigated for coherent evolution dynamics using DL methods and advanced characterization[45–47], the use of deep learning for studies of incoherent nonlinear optical processes has been the subject of only limited study[48–51]. Neural networks have indeed demonstrated the ability to capture and analyze incoherent pulse formation in fibres[48,49], or to predict the formation of chaotic pulses in Kerr resonators from the analysis of time-series precursors[51]. These results however rely on output waveform analysis (i.e. temporal time series or spectral shot-to-shot fluctuations), but do not provide access to actively tunable initial conditions nor take into consideration a more macroscopic characterization of the dynamical system. The real potential of DL for controlling incoherent signals and their fluctuation properties thus remains largely unexplored, despite its fundamental interest and its strong applicative potential in many areas employing incoherent optical waveforms.

Among numerous incoherent processes encountered in photonics, Modulation Instability (MI) is perhaps the canonical example of a physical phenomenon that bundles both complexity and noise-sensitive dynamics[52,53]. In this work, we consider this key process in which weakly modulated signals undergo amplification, leading to the generation of new frequency components[54,55]. In nonlinear optics, MI arises from the interplay between Kerr nonlinearity and (anomalous) dispersion[53] and, when spontaneously generated from noise, leads to complex and rich interactions yielding to the emergence of randomly localized structures[53,56]. As such, for over 15 years, MI has served as the ideal system for the fundamental study of extreme event formation in optics, while providing an excellent testbed for real-time characterization techniques of incoherent processes[47,56–61]. MI has also been a focal point of several studies aiming to leverage ML techniques for optical processing, and, for the control of coherent MI dynamics[62,63], several works have studied the phase space topology of idealized Four Wave Mixing (FWM)[41,64,65]. However, incoherent MI generated from noise offers only limited control of propagation dynamics, whether through optical seeding[66–68], modulation[69] or fibre engineering[70]. Moreover, these approaches are either supported by limited theoretical frameworks[71,72] or lack the capability to explore a wider parameter space in experiment. For a practical comparison of the advantages and limitations of ML compared to standard numerical modelling or classical signal processing techniques, one can refer to recent reviews on the topic[8,9,18,19,26,27,29]. In the Supplementary Information, we also provide a succinct overview of the impact of ML methods on selected applications relevant to nonlinear pulse propagation in guided wave optics (see Table S1). However, we note that paired with the practical constraints associated with extended dataset acquisition of specifically tailored and dynamically reconfigurable incoherent propagation dynamics, ML was merely exploited for studying MI and engineering noise-driven spectral broadening in an experimental context.

In this work, we investigate—both numerically and experimentally—whether artificial neural networks can be suitably leveraged to forecast the output properties of such incoherent spectral processes and their underlying initial conditions. Specifically, we explore the potential of a coherent seed control to actively manipulate incoherent spectral broadening during nonlinear fibre propagation. In particular, we discuss the potential of optical seeding, paired with deep learning, to reshape and tailor specific correlation features in the broadband output signal, towards harnessing ML for incoherent and nonlinear optical information processing.

## Results

The principle of our study is illustrated in Fig.1, introducing a simple yet efficient approach to leverage coherent optical seeds to mitigate and tune incoherent dynamical evolution arising from concurrent noisy processes[53,73].

Spontaneous MI, occurring when a coherent pulse copropagates with an incoherent noise field in a nonlinear fibre (Fig. 1a), triggers broadband noise amplification and eventual cascaded four-wave mixing processes. The resulting output spectrum exhibits a clear broadening with large shot-to-shot fluctuations emerging from different input noise conditions (see Methods). In the temporal domain, the formation of highly-localized pulse structures can be observed, which cannot be seen when averaging the signal over multiple realizations (red lines). However, adding weak but coherent optical signal(s) to the noisy initial pulse (Fig. 1b) has a significant impact on the propagation dynamics and allows for a relative control of the MI spectral broadening processes. In this so-called seeded MI scenario, the nonlinear evolution remains incoherent, so that the output waveform still exhibits significant shot-to-shot fluctuations. Yet, a noticeable structuring can be observed when averaging the fluctuating spectral and temporal output profiles (see also Supplementary Figs. S1 and S2). Here, we specifically assess the capability of an ANN to train on extensive numerical and experimental datasets for predicting such noise-driven propagation features (i.e. predicting output average spectra and correlation properties).

Our ANN approach, illustrated in Fig. 2, encompasses both forward propagation (predicting incoherent dynamics from initial seeding conditions) and backward propagation (inferring seeding input conditions from output features). Both approaches rely on the same 5-layer neural network architecture, serving as a standardized and reproducible framework, with input and output layers respectively fed with either input seeding conditions or output spectral features (for either ANN inferences or predictions).

Below, we validate our approach through both numerical simulations and experiments of tailored MI field evolution in fibres characterized via real-time spectral measurements (see Methods), therefore assessing the efficiency of neural networks in enhancing control of complex nonlinear optical systems in realistic conditions. We first demonstrate that this ANN-based method enables retrieving the input seed parameters—wavelength and phase—with accuracy despite the challenges posed by weak nonlinear signal amplification and complex noise-mediated evolution dynamics. We then benchmark the network's forecasting capability for the prediction of spectral correlation maps. Finally, we study and discuss the aptitude of the network to optimize specific correlation features, further demonstrating its practical utility for tailoring incoherent waveforms properties highly sought after in many areas of photonics (spanning e.g. incoherent imaging, compressed sensing, as well as classical and quantum signal processing, to only name a few).

### Forecast of incoherent nonlinear dynamics from artificial neural networks

For the numerical prediction of incoherent dynamics, we generated an extensive numerical dataset to train the ANN by performing Monte-Carlo simulations of fibre pulse propagation across ~100,000 seeding scenarios (see Methods). These simulations, based on the generalized nonlinear Schrödinger equation[74] (GNLSE), consider the propagation of a 29.1 ps optical pulse at 1560 nm through a 485 m highly nonlinear fibre (HNLF). Such a case is associated with nonlinear fibre propagation in a weakly anomalous dispersion regime, where MI occurs spontaneously from broadband noise amplification (Fig. 1), and it corresponds to typical parameters used in our experiments.

In our simulations, for each realization, we consider different broadband white noise as well as amplified spontaneous emission (ASE) noise, filtered over a 5.2 nm bandwidth to cover most of the red-

detuned MI gain (see Methods). This random ASE noise–mimicking the experimental noise associated with an Erbium-doped fibre amplifier (EDFA)–is relatively weak compared to the excitation field (−60 dB level, see Fig. 1a) but is progressively amplified during propagation until driving spectral broadening via cascaded MI processes. When considering a seeded MI configuration (see Fig. 1b), two (or four) coherent optical seeds $C_i(\lambda_i, \varphi_i)$ are also injected with the input pump pulse. For each seeding configuration, these very weak seeds (i.e. −50 dB compared to the pump) exhibit arbitrary phases $\varphi_i$ and wavelengths $\lambda_i$ selected within the same bandwidth as the ASE noise (i.e. in the range 1562.5−1564.6 nm, see Methods for details).

To assess the impact of the coherent seeds $C_i$ on the overall evolution dynamics, we first computed, for each input seed configuration, the nonlinear propagation of the pulse over 500 noisy realizations. As illustrated in Fig. 2, the shot-to-shot fluctuations of the output spectra are then analyzed to retrieve the average spectrum $S(\lambda)$ as well as the spectral intensity correlation map $\rho(\lambda, \lambda)$ via a linear Pearson metric[47,57]. This straightforward approach allows for characterizing the interplay of broadening dynamics resulting from the competition between coherent seed amplification and noise-driven MI processes, here acting at a comparable level (see Methods). More importantly, such a method allows for easily converting over 50 million numerically generated incoherent spectra (i.e. -100,000 seed configurations with 500 noisy realizations each) into a large dataset of steady (i.e. stationary) statistical metrics that can be readily exploited for ANN training. Indeed, here, we numerically investigate either 2-seed and 4-seed scenarios for 90,000 and 105,000 seed configurations respectively, generating an extensive dataset of output spectral properties $S_n(\lambda)$ and $\rho_n(\lambda, \lambda)$ depending on random input seed parameters $C_n$ (with e.g. $C_{n,1}(\lambda_{n,1}, \varphi_{n,1})$, $C_{n,2}(\lambda_{n,2}, \varphi_{n,2})$ for each 2-seed configuration).

For ANN training, we consider a rather simple fully-connected Feed-Forward Neural Network (FFNN) model trained with -100,000 different seeding configurations. For both ANN inferences and predictions, we use a similar FFNN featuring five layers and a total number of individual neurons (perceptrons) ranging from 1676 to 37,392 depending on the FFNN conformation (see Table S4 in the Supplementary Information). Training is achieved in 40 minutes for the less demanding tasks, whilst in up to almost 10 hours for the largest neural networks, trained in conservative fashion (see details in Table S5). In all cases, the ANN accuracy is tested against a set of previously unseen configurations by the network (see Methods). As a numerical benchmark, we first train our ANN to predict the average output spectra $S_n(\lambda)$ resulting from the incoherent broadening of arbitrary input seed parameters $C_n(\lambda_1, \varphi_1, \lambda_2, \varphi_2)$. In a second step, we consider the ANN's ability to infer $C_n$ seed properties such as wavelengths $\lambda_1$ and $\lambda_2$ and phase difference $\Delta\varphi_{2,1} = \varphi_2 - \varphi_1$, from the correlation map $\rho_n$ of associated spectral fluctuations after noisy nonlinear amplification. For both 2-seed and 4-seed dataset, the results of ANN training from numerical simulations are summarized in Fig. 3.

Figure 3a. shows an excellent agreement between the ANN predictions and the simulation results when forecasting the average output spectrum from dual-seeding propagation scenarios. These predictions showcase a great accuracy over a 50 dB dynamic range with as little as 1.0% Root Mean Square Error (RMSE) compared to GNLSE simulations.

More importantly, Fig. 3b highlights the ANN's potential to successfully retrieve the seed properties from the statistical analysis of noise-mediated spectral broadening. From the correlation maps $\rho_n$, the network demonstrates the ability to infer seed parameters with an average error below 2.4% RMSE. These inference results are shown in Fig. 3b though a density plot of the ANN prediction datapoints (white

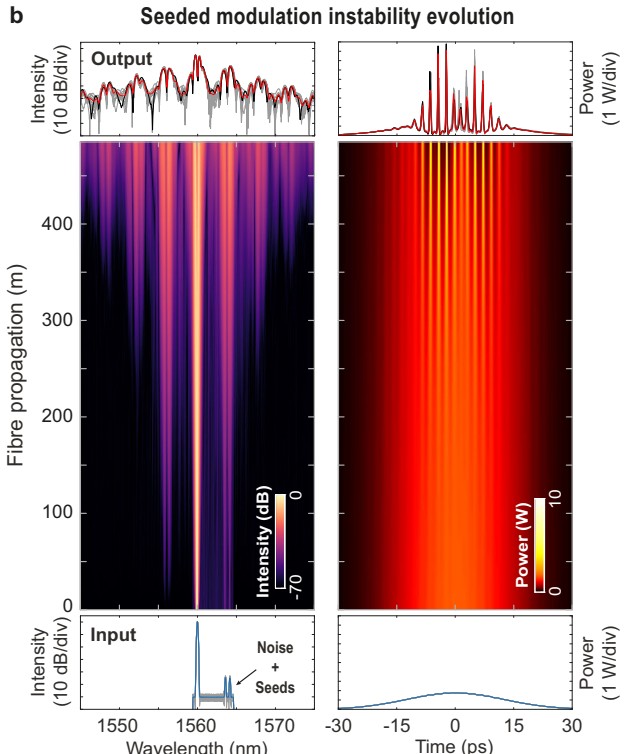

**Fig. 1 | Illustration of incoherent modulation instability dynamics.**
**a** Spontaneous modulation instability dynamics: Spectro-temporal evolution of noisy input pulse (29.1 ps) propagating nonlinearly in an optical fibre, obtained via numerical simulations. **b** Seeded modulation instability dynamics: Spectro-temporal evolution of the same noisy input pulse copropagating with two weak but coherent optical seeds temporally superimposed with the pump. In both cases, the averaged input and output intensity profiles are shown with thick blue and red lines, respectively. The results of different noisy realizations are shown with dashed grey lines while the intensity profiles of the selected case (displayed in the evolution plot) are shown with a black line.

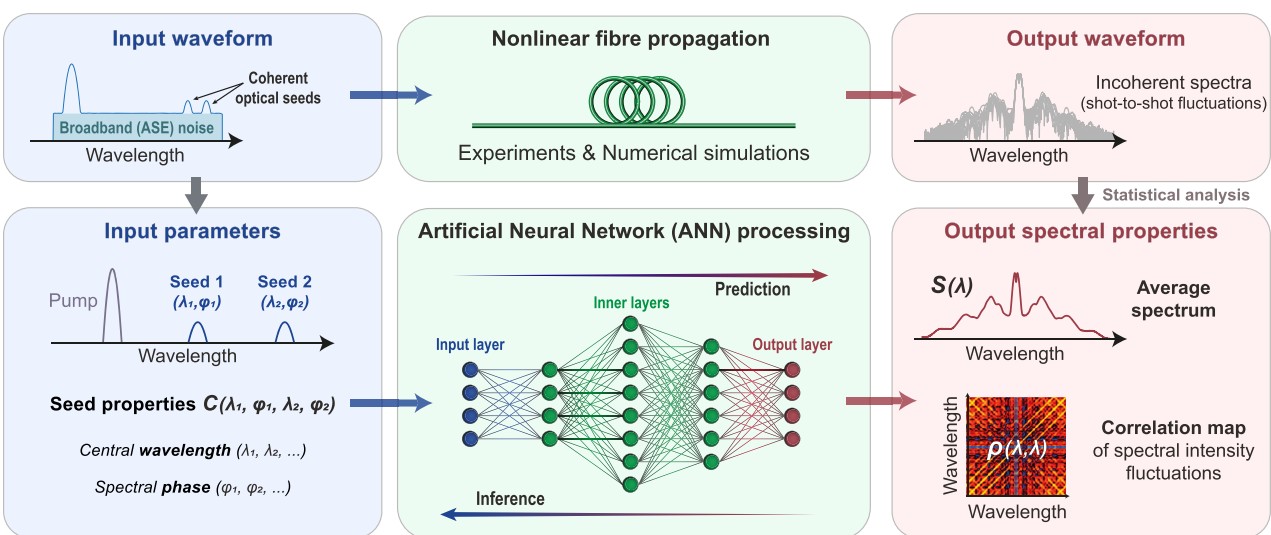

**Fig. 2 | Deep-learning strategies for the forecast of noise-driven nonlinear instabilities.** Schematic of an artificial neural network (ANN) architecture for the analysis of incoherent modulation instability dynamics. The ANN is employed to predict the output spectrum and fluctuation properties of seeded modulation instability dynamics depending on the input seed parameters. Similarly, the properties of the fluctuating output spectrum (i.e. average spectrum and spectral correlation maps) can be used to infer the input seeds parameters. Both these ANN approaches can be compared to numerical simulations and experimental results considering incoherent modulation instability dynamics in nonlinear fibre propagation (see Methods).

dots) in excellent agreement with the ground truth obtained from GNLSE-based simulations (dashed blue line).

It is worth noting that for the seeded MI regime considered here, the impact of noise (and the seed phase) is paramount, so that the input seed locations do not readily correspond to the regions of maximum correlation or anticorrelation within the output spectrum (nor the maxima in the average output spectrum - see Supplementary Figs S1 and S2). Yet, in this case, the seed wavelengths are inferred with less than 2.1% error (RMSE), which corresponds to a spectral precision of 42 pm.

In fact, the relative phase between the seeds is crucial in the four-wave mixing processes involved in the MI dynamics at play during nonlinear fibre propagation. However, the phase is not a parameter readily observable through a simple spectral intensity analysis. More importantly, the weak relative power of the seeds (i.e. −50 dB) and the presence of broadband noise yield an input signal with a very low modulation depth (see Fig. 1b) that can make phase retrieval a challenging task in experiments. Yet, through noise-driven nonlinear amplification, the analysis of the output spectral fluctuations from the ANN allows inferring the relative phase difference between the seeds with very good accuracy (i.e. 3.1% RMSE−see Fig. 3b).

In essence, this approach not only demonstrates the ANN's capacity in forecasting nonlinear fibre dynamics but also emphasizes its potential in decoding complex and noisy nonlinear interactions in fibre optics. For instance, extending to the approach to the 4-seed case, Fig. 3c confirms the ANN's proficiency in predicting output spectral profiles with an average RMSE below 2.9% over the same dynamic range. Similarly, Fig. 3d displays how the inference of input seed parameters remains very efficient even within an extended parameter space, achieving an error comprised between 3.0% and 8.5% for the retrieval of the four input seed properties.

**Experimental validation of incoherent nonlinear dynamics inference via ANNs**

Following the numerical proof of concept above, we proceed to an experimental validation of our deep-learning approach for the ANN prediction and inference of noise-driven MI dynamics. The experimental setup, depicted in Fig. 4, allows us to acquire extensive experimental datasets to assess the impact of coherent optical seeding on incoherent MI broadening dynamics (see Methods for details).

Starting from a broadband signal in the C-band (i.e. ~ 80 fs pulse centred at 1560 nm), we filter out a ~ 12 ps pump signal at 1556.8 nm, as well as two (or four) coherent optical seeds with adjustable properties. Specifically, the picosecond pump and weak coherent seeds (at −25 dB) are both generated from a programable Fourier optical filter[57] (Waveshaper−WS) allowing to readily tune the seed parameters (i.e. $C_n$ with random wavelengths and phase) in the 5 nm spectral range spanning 1551.3–1556.3 nm. Once generated, this input signal is amplified via an erbium-doped fibre amplifier (EDFA) to reach a suitable peak power while it also provides additional amplified spontaneous emission (ASE) noise to the input pulse. To adjust the relative impact of this broadband noise (and its competition with the coherent seeds) on the subsequent propagation dynamics, the ASE is filtered to match the spectral bandwidth of the blue-detuned MI gain during pulse propagation (see Methods). This prepared input signal is then injected into 385 m of HNLF where it undergoes spectral broadening along the dynamics depicted qualitatively in Fig. 1b.

At the fibre output, the average spectrum can be directly detected by an optical spectrum analyzer (OSA) while the shot-to-shot spectral fluctuations are measured via a time-stretched dispersive Fourier transform (DFT) to monitor real-time instabilities in the incoherent output spectrum[47]. While further details are provided in the Methods section, it is worth noting that the DFT technique is here implemented to provide optimal spectral resolution (0.48 nm resolution, with an equivalent sampling rate of 0.12 nm/point) while avoiding artefacts associated with temporal signal overlap, dynamic range limitations and detector impulse response. To this end, the incoherent spectrum is filtered via another programable Fourier optical filter to perform real-time spectral measurement on only the red-detuned MI sideband of the output signal (i.e. > 1556.8 nm). DFT is implemented via a long section of dispersion compensating fibre (DCF) coupled to a 5 GHz bandwidth detection architecture (i.e. photodiode and real-time oscilloscope). These shot-to-shot temporal traces can then be leveraged to reconstruct the spectral correlation map $\rho_n(\lambda, \lambda)$ over the range 1557–1583 nm (using 1000 DFT traces per seed setting $C_n$), but also to retrieve the average spectrum $S_n(\lambda)$ computed over the same region of interest.

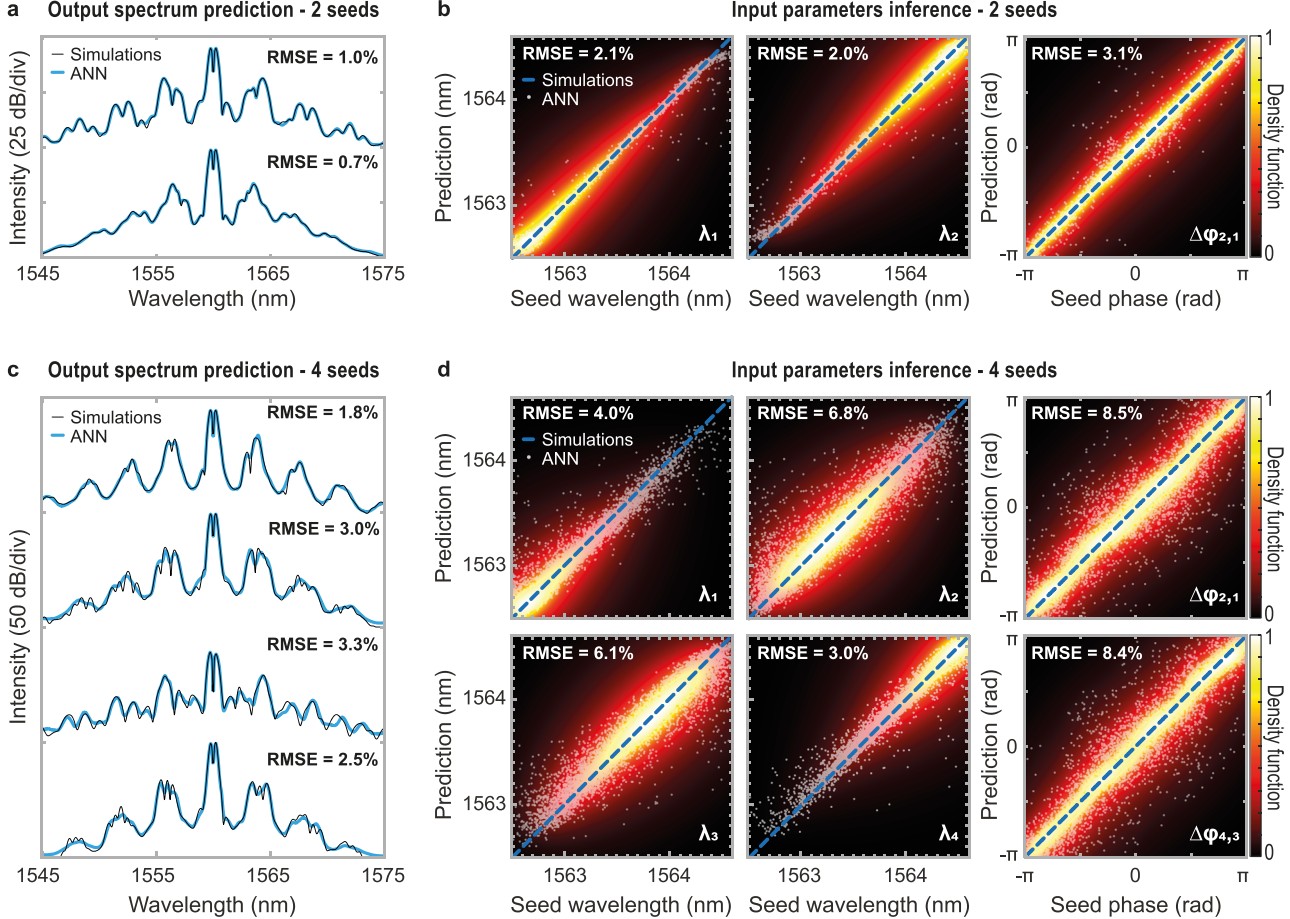

**Fig. 3 | Prediction and inference of noise-driven modulation instability dynamics obtained from numerical simulations. a** Example of average spectra obtained after fibre propagation for different dual-seed configurations at the fibre input. The artificial neural network (ANN) prediction (thick blue line) is superimposed on the ground truth given by the simulations (thin black line). **b** Two-seed wavelength and phase parameters ($\lambda_1$, $\lambda_2$ and $\Delta\varphi_{2,1}$) retrieved by the neural network (white dots) and compared to the seed parameters used in simulations (dashed blue line). The dispersion of the ANN predictions compared to the ground truth (i.e. simulations parameters) is also illustrated via a density plot of the network predictions in each panel background and the root mean square error (RMSE) is provided for each ANN prediction. **c** Example of average output spectra obtained for different four-seed configurations at the fibre input. Other parameters are the same as in panel (**a**). **d** Four-seed wavelength and phase parameters ($\lambda_1$, $\lambda_2$, $\lambda_3$, $\lambda_4$, $\Delta\varphi_{2,1}$ and $\Delta\varphi_{4,3}$) retrieved by the neural network (white dots) and compared to the seed parameters used in simulations (dashed blue line). Other parameters are the same as in panel (**b**).

The setup of Fig. 4 is leveraged to perform data acquisition over a large range of seeding parameters: for each seeding scenario (i.e. either 2 or 4 seed inputs), an experimental dataset comprising 60,000 random seed configurations $C_n$ is used to train our ANNs (see Methods). Here, we start by training the networks to infer the input seed parameters $C_n$ from the output correlation map $\rho_n(\lambda, \lambda)$, using a 5-layer FFNN with up to ~28,000 perceptrons (see Table S4). Conversely, similar ANNs are also trained to forecast the output spectrum $S_n(\lambda)$ either from the input seed parameters $C_n$, or from the output correlation map $\rho_n(\lambda, \lambda)$. In each case, we used the same FFNN architecture, with 5 layers and adjusted layer widths depending on the type (i.e. vector length) of datasets used as input and output (see Methods).

The ANN results considering dual-seeding are summarized in Fig. 5a, b. Even in such a noisy experimental regime, where the weak seeds are dynamically competing with the ASE during incoherent spectral broadening, the network can readily infer the input seed parameters (i.e. $\lambda_1$, $\lambda_2$ and $\Delta\varphi_{2,1}$) with good accuracy. Considering the validation set only (i.e. 5% of the total dataset, 3000 different correlation maps and seeding cases unused during training), the ANN yield an error (RMSE) below 5.0% for the wavelength inference, and down to 7.3% for the phase retrieval (see Fig. 5b).

Interestingly, the prediction of the average spectrum from the input seed parameters is not as accurate. While the RMSE remains relatively low, around 4.1%, such a prediction scheme is unable to reproduce the fine structure of the MI sidebands in e.g. three representative spectra displayed in Fig. 5a (orange dashes). However, using the same ANN scheme but with the correlation maps as the input layer of the ANN, the network can reach an average error below 1% RMSE (see thick blue lines), with an excellent agreement with ground truth obtained from averaged DFT time traces (black lines).

A qualitatively similar behaviour can be obtained when considering a 4-seed input case, whose experimental results are summarized in Fig. 5c, d. The inference of the seed parameters, here illustrated for both wavelengths and phases ($\lambda_1$, $\lambda_2$, $\lambda_3$, $\lambda_4$, $\Delta\varphi_{2,1}$ and $\Delta\varphi_{4,3}$), is in general agreement with the experiments but shows some limitations as the average prediction error reaches 15.4% for the wavelengths and 24.2% for all six combinations of phase difference $\Delta\varphi$ between the four input seeds.

Figure 5c also includes three illustrative spectra predictions from the ANNs: the network yields an average error of 2.8% for the prediction arising from the seed parameters and 0.8% for the prediction based on the correlation maps.

The ANN results obtained for the 4-seed experimental dataset are consistent with both numerical results and experimental results using fewer seeds. Although the ANN accuracy levels shown in Fig. 5d may seem modest−especially for phase inference−they highlight the

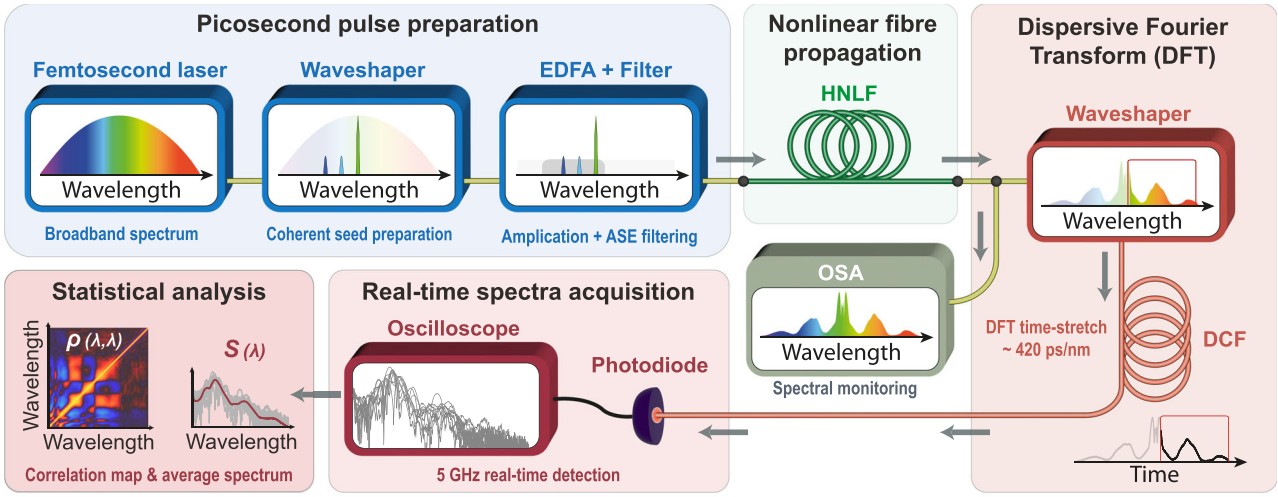

**Fig. 4 | Experimental setup.** Schematic of the experimental setup implemented for reconfigurable modulation instability seeding and real-time spectral characterization. The data obtained from dispersive Fourier transform (DFT) signal processing and real-time optoelectronic acquisition can be leveraged for artificial neural network training after suitable statistical analysis. (EDFA: Erbium doped fibre amplifier; ASE: amplified spontaneous emission; HNLF: highly-nonlinear fibre; DCF: dispersion compensating fibre; OSA: optical spectrum analyzer).

increased difficulty of inferring parameters from a larger set of seeds, where errors are 2.5 to 3 times higher compared to the dual-seed scenario (a feature also observed in seed parameters inferred from numerical data, see e.g. Fig. 3b, d). Moreover, the ANN inference of seed parameters from experimental data shows approximately a threefold degradation in accuracy relative to numerical data, likely due to experimental instabilities such as weak EDFA power fluctuations during data acquisition (see Supplementary Figs. S3 and S4). As such, these results illustrate the stability limitations associated with our experimental system. Nevertheless, these also demonstrate that the signature of incoherent fluctuations proves useful to retrieve weak input parameters via inherently complex and noisy nonlinear propagation dynamics. In the next section, we explore whether ANN approaches can be instead leveraged to mitigate time-consuming noisy simulations and complex real-time measurements in order to study incoherent MI spectral broadening dynamics.

## ANN prediction of incoherent spectral fluctuations

The assessment of the ANN capability to predict the statistical shot-to-shot fluctuation properties within the output spectrum is here conducted both numerically and experimentally, leveraging the same 5-layer FFNN with adjusted layer widths (see Methods and Table S4). First, using numerical simulations, we train our network to predict the shape of the spectral correlation map $\rho_n(\lambda, \lambda)$ from the seed parameters. These results, considering either two or four seed datasets, are summarized in Fig. 6.

Figure 6a shows two representative examples of the network's correlation map prediction over the whole spectral range (i.e. 1540–1580 nm), here obtained considering different dual seeding scenario. The average prediction error over the ANN validation set is 3.7% (RMSE), thus showing a very good agreement between the network's prediction and the simulation results.

Similarly, for the four seed case, Fig. 6b illustrates examples of the network correlation map predictions. In this regime, we also observe a good quantitative agreement with the GNLSE simulation results, so that the ANN average error (RMSE) remains at 7.4% for the validation set of the 4-seed scenario.

Importantly, the correlation maps presented in Fig. 6 exhibit strong signatures that result from considering close-to-ideal input noise properties prior to nonlinear fibre propagation (i.e. broadband quantum noise and low-amplitude filtered ASE noise—see Methods). In

our simulations, the nonlinear amplification of the weak coherent seeds and their competition with noise yield well-defined and reproducible spectral fluctuations (see Fig. 1b). It is therefore possible for the network to predict these correlation signatures with accuracy, despite the chaotic nature of the dynamics involved in such propagation phenomena.

In the experimental realm, the lack of control over fine-tuned experimental conditions and the intrinsic sensitivity and variability of the nonlinear system to the initial conditions make the acquisition of correlation maps with similar dynamics -and correlation signatures- quite challenging. However, this is typically in such a complex regime, falling outside the scope of viable analytical tools and realistic numerical approaches, that deep learning techniques can prove extremely useful.

To assess this potential, we used the experimental datasets, obtained via the DFT setup shown in Fig. 4, to train neural networks towards the reconstruction of spectral correlation maps for different seeding conditions. The results of ANN predictions are summarized in Fig. 7, with examples illustrated for cases from both 2 and 4 seed scenarios.

Figure 7a illustrates the network's performance in predicting experimental correlation maps for two examples of the 2-seed MI propagation scenario. The experimental maps $\rho_n$ (top row) are compared with ANN map predictions considering either the mean output spectrum $S_n$ (middle row) or the input seed parameters $C_n$ (bottom row) as the network input parameters.

For this dual seeding scenario, one can see that the network's prediction derived from the input parameters provides a reasonably good qualitative agreement with the experiments and the ANN can forecast the predominant correlation signatures in the maps. Yet, the fine structures are not perfectly reproduced, and the prediction errors (RMSE) remain relatively high (8.1% average error for the 3000 maps of the validation dataset).

Interestingly, the correlation maps predicted from the ANN trained on the mean spectral trace yield better qualitative agreement with the DFT experiment, with spectral correlation signatures that are well reproduced and a drastically reduced prediction error of 4.0% (RMSE) averaged over the whole validation dataset.

It is worth noting that this observation still holds when considering a larger tunable parameter space, as provided with 4-seed scenario depicted in Fig. 7b. ANN correlation map forecast from the

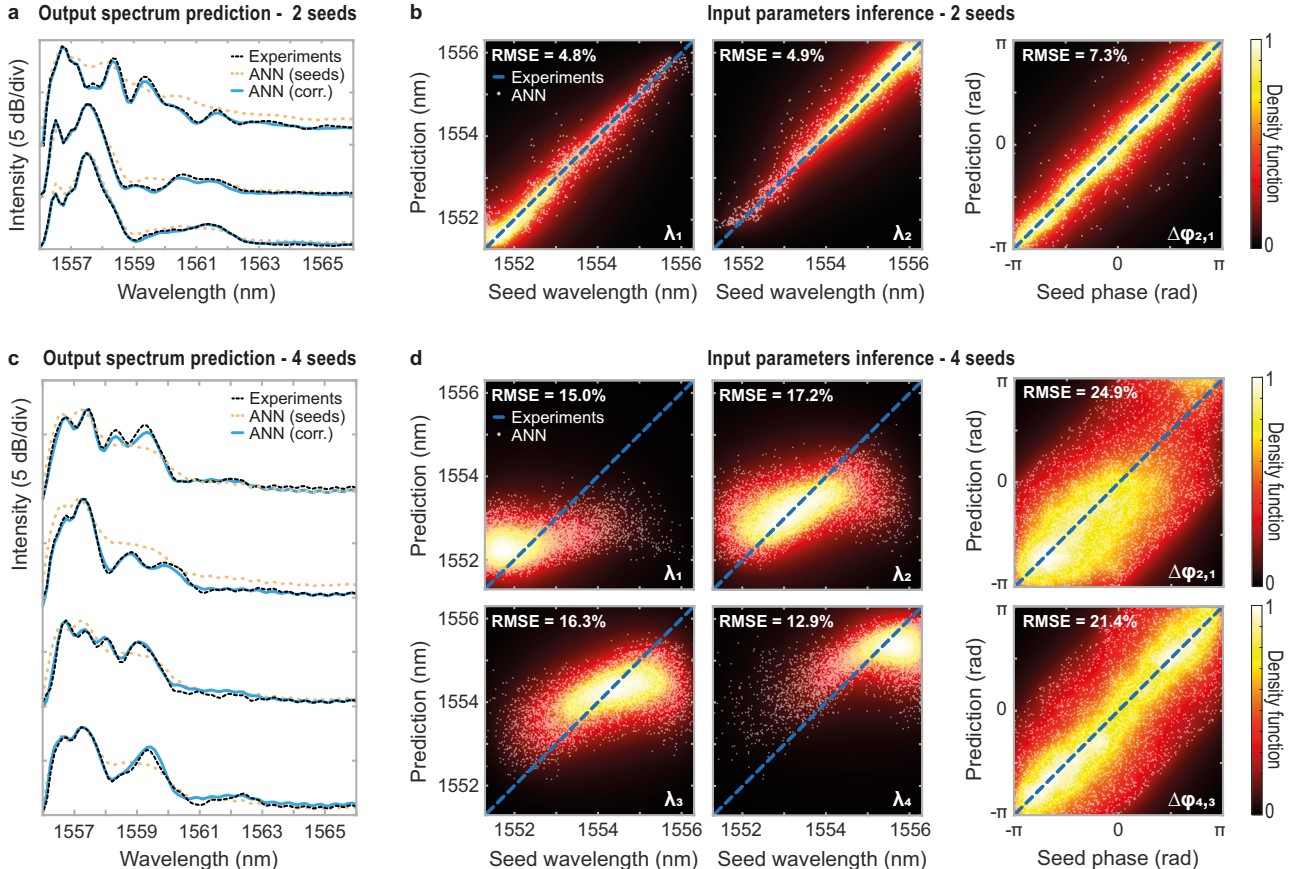

**Fig. 5 | Prediction and inference of experimentally measured noise-driven modulation instability dynamics. a** Example of three average spectra obtained after fibre propagation for different two-seed configurations at the fibre input. The artificial neural network (ANN) predictions obtained from experimental correlation maps (thick blue line) and from input seed parameters (orange dashes) are superimposed on the ground truth given by the experimental measurements (dashed black line). **b** Two-seed wavelength and phase parameters ($\lambda_1, \lambda_2$ and $\Delta\varphi_{2,1}$) retrieved by the neural network (white dots) and compared to the seed parameters used in experiments (dashed blue line). The dispersion of the ANN predictions compared to the ground truth (i.e. experimental parameters) is also illustrated via a density plot of the network predictions in each panel background and the root mean square error (RMSE) is provided for each ANN prediction. **c** Example of four average spectra obtained after fibre propagation for different four-seed configurations at the fibre input. The display parameters in panels (**a**) and (**c**) (respectively associated to the two and four seed configurations) are the same. **d** Four-seed wavelength and phase parameters ($\lambda_1, \lambda_2, \lambda_3, \lambda_4, \Delta\varphi_{2,1}$ and $\Delta\varphi_{4,3}$) retrieved by the neural network (white dots) and compared to the seed parameters used in experiments (dashed blue line). The display parameters in panel (**b**) and (**d**) (respectively associated to the two and four seed configurations) are the same.

average spectrum $S_n$ yields better results than an ANN trained from the input seed parameters $C_n$. Yet, in both cases, the agreement with experiments is better, with an average prediction error (RMSE) within the network validation dataset that decreases slightly, respectively reaching 3.9% for predictions arising from the output spectral trace, and 7.6% for predictions arising from the input parameters.

Overall, the results summarized in Fig. 7 highlight the potential of our neural network to successfully forecast incoherent signal correlation properties, based on tailored initial conditions or from specific (average) output spectral intensity signatures. These results further underpin the capability of deep-learning techniques to be suitably applied in a realistic experimental condition, taking advantage of nonlinear noise amplification and competition between different and complex nonlinear wave-mixing processes with various coherence degrees.

From a more specific analysis of these ANN prediction results, the conclusions are two-fold: First, the forecast of the correlation maps appears to be more effective when considering a larger parameter space in the initial conditions (i.e. 2-seed vs 4-seed scenario). While not unexpected, as ANNs typically require a large variability for successful training, the inherent system complexity does not seem to be limiting (at least for a reasonably low dimensionality of the input parameter space) but rather allows for generating differentiating signatures that the network can leverage for a better prediction (for a similar ANN architecture).

Second, utilizing output spectral signatures for experimental predictions has proven more effective and reliable in capturing complex correlation features than relying solely on the input seed parameters. In contrast with ANN predictions based on numerical simulations (see Fig. 6), these findings underscore the limitations of such a deep learning technique within an experimental system highly sensitive to the initial conditions (e.g. type of noise). In particular, experimental data analysis shows that the predominant cause of unpredictability in our system seems to be associated to the long-term instability of our fibre amplifier (EDFA) during large dataset acquisitions (see Figs. S3 and S4 in the Supplementary Information).

While we expect these limitations to be potentially mitigated, together, these points raise the open question of uniqueness, reproducibility, complexity and potential generalization of the incoherent nonlinear dynamics within the system at play (and whether this chaotic system can be effectively modeled or forecast by its initial conditions alone in an experimental environment).

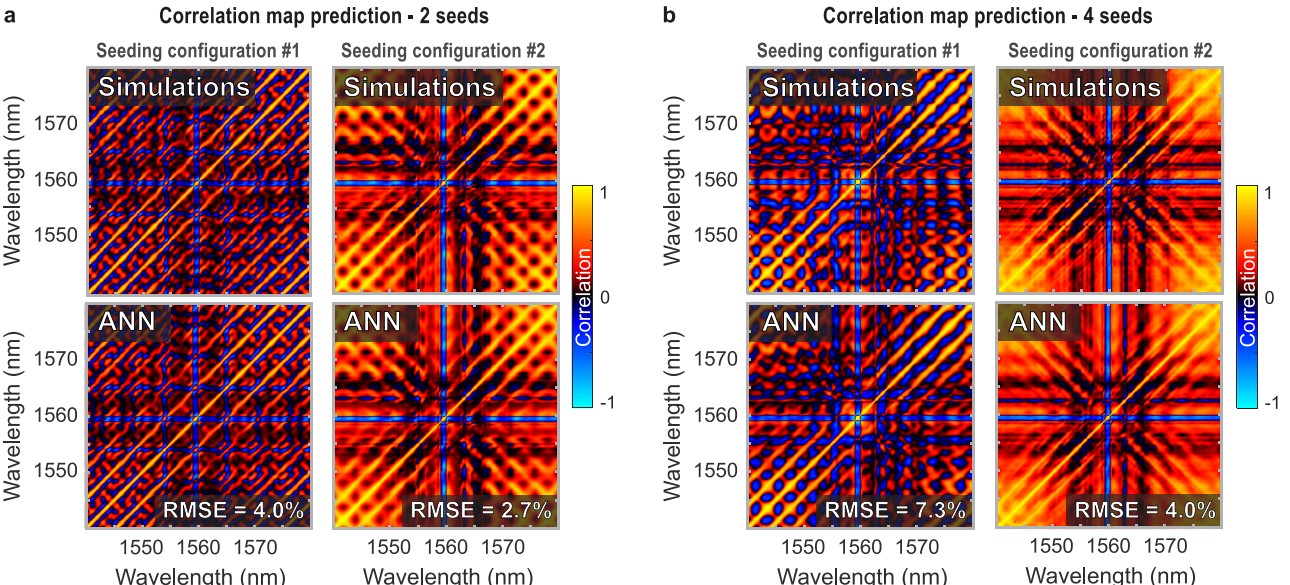

**Fig. 6 | Prediction of spectral correlation maps by artificial neural networks trained from numerical simulations. a** Example of two correlation maps predicted by the artificial neural network (ANN) in the dual-seed case (bottom), corresponding to two representative yet different seeding conditions. The corresponding correlation maps computed from numerical simulations are displayed on the top panel. **b** Same as panel (**a**), illustrating two examples of correlation maps predicted by the ANN in the four-seed case. In all cases, the root mean square error (RMSE) value displayed is computed as the average of each pixel RMSE over the whole map.

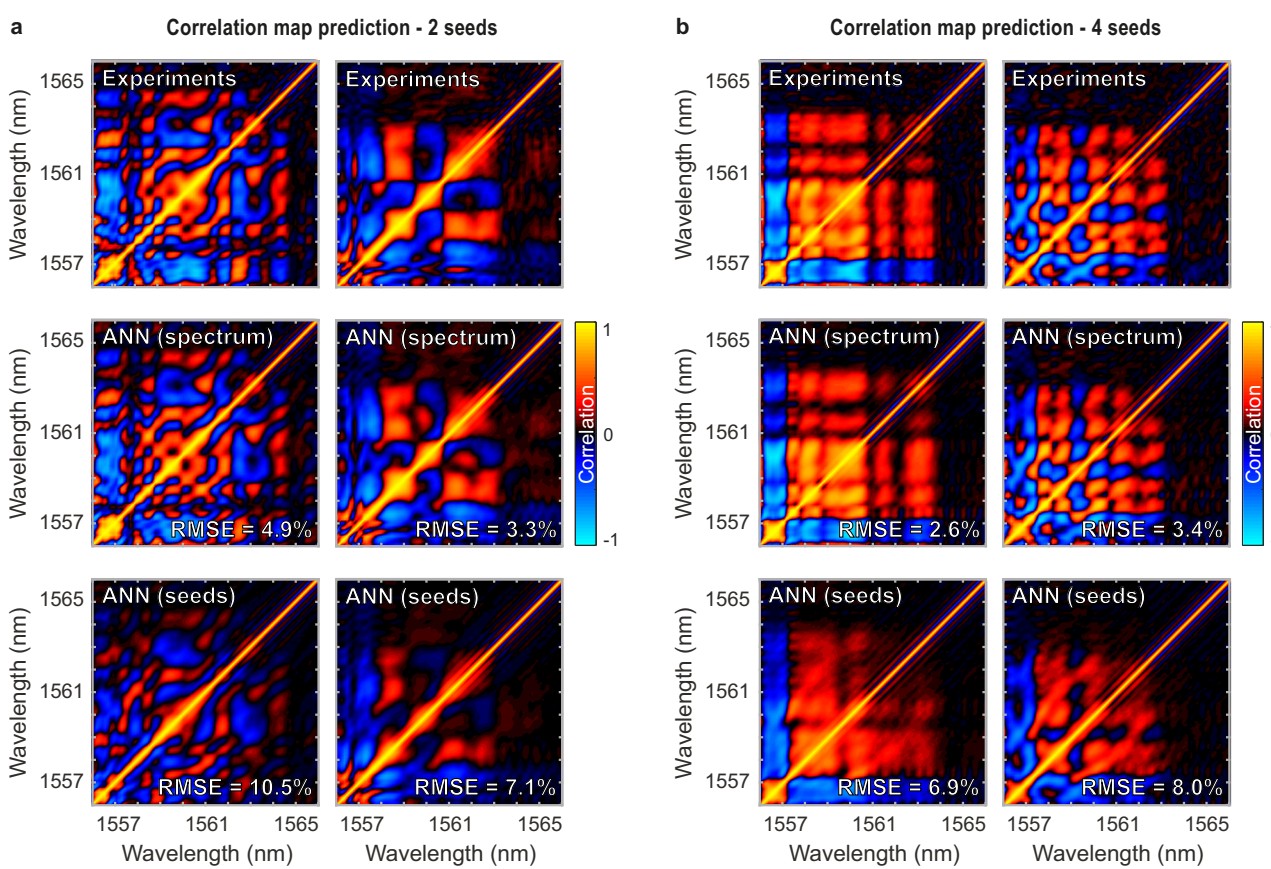

**Fig. 7 | Prediction of spectral correlation maps by artificial neural networks trained from experimental datasets. a** Two-seed experiment: Example of two experimental correlation maps predicted from the neural network for different seeding conditions. The correlation maps predicted by the artificial neural network (ANN) trained either from the input seed parameters (bottom panels) or from the output spectrum (middle panels), are compared to the correlation maps retrieved from experiments displayed on the bottom panel (top panels). **b** Same as panel (**a**), illustrating two examples of correlation maps predicted by the ANN for experiments using four input seeds. In all cases, the root mean square error (RMSE) value displayed is computed as the average of each pixel RMSE over the whole map.

## Discussion

In this study, we have demonstrated numerically and experimentally that ANNs can effectively address both inference and prediction problems when considering tunable incoherent nonlinear processes. In fact, our approach proved to be suitable for predicting nonlinear dynamical features such as correlation maps and for uncovering hidden information concealed in noisy dynamics. Specifically, in two-seed scenarios, the network can infer the input parameters with an error below 8% in all cases. Yet, this error increases with the number of seeds considered. Conversely, for the prediction of output features, the ANN maintains correlation map prediction errors below 9% for both two-seed and four-seed configurations. In this case, a degradation of the network predictions is however observed in experiments when training the ANN from input seed conditions rather than average output spectral measurements.

From a global viewpoint, we note that the proposed methodology can significantly enhance optimization time and data generation costs. In this study, we demonstrated the efficacy of a feedforward neural network (FFNN) with a fully-connected conformation, for a proof-of-concept demonstration relying on a simple and reproducible ANN architecture - here serving as a robust benchmark to guarantee a certain degree of universality in the trained network (i.e. varying only in the dimensionality of ANN input, inner, and output layers to adapt to different inference and prediction scenarios). We estimate, however, that further enhancements in the ANN's architecture and training parameters could improve predictions and increase the reliability of this optimization method in both theoretical and experimental setups. For instance, we expect that such promising results can be further improved with advanced ANN frameworks (e.g. exploiting refined Python-based toolboxes for in-depth studies focusing on specific network architectures and conformations). From an experimental perspective, leveraging better detection features (e.g. improved DFT dynamic range, sensitivity, resolution) and overall long-term stability in our experiments is also expected to provide a better quantitative agreement for ANN training.

In this study, as in most works involving deep learning, we found that the accuracy of ANN-based training and prediction is indeed largely dictated by the data quality. Yet, there are clear opportunities for improvements in terms of network architecture and hyperparameter tuning, depending on whether the focus is on fundamental studies (leveraging e.g. physics informed networks) or applied aspects relying on e.g. online optimization (metaheuristic search, reinforcement learning strategies, etc.).

For instance, we expect that tailored ML techniques and ANN architectures might be better suited to different types of input-output data[18,19]: autoencoder approaches may be relevant for ANN training from initial seeds (i.e. predictions), while Convolutional Neural Networks (CNNs) should be naturally adapted for training from correlation maps (i.e. treating them via image processing tools and pattern recognition techniques). Early studies on CNN architectures achieved moderate success (similar to FFNN) but could be further explored in subsequent works. Once refined, these methods may improve accuracy or speed, depending on the datatype considered. More advanced architectures such as Generative Adversarial Networks (GANs) also hold further promise for optimizing both prediction and inference models in a unified framework[75].

Preliminary work also showed that simple evolutionary algorithm-based metaheuristics (e.g., genetic algorithms or particle swarm optimization) can iteratively optimize MI correlation features in noisy and realistic experimental conditions[76], although each target function still needs its own specific optimization (i.e. no training nor generalization possible). Along this line, Reinforcement Learning (RL) shows strong potential for online optimization of noisy and dynamically changing experimental conditions within complex, nonlinear and chaotic systems[77], as recently demonstrated numerically for suppressing MI in

spatial beams[78]. Following recent advances in other fields of photonics[8,9,18,19], deep RL clearly constitutes an excellent strategy for mitigating changes in environmentally unstable conditions (e.g. to consider and mitigate both thermal drifts and EDFA instabilities in realistic experimental conditions), while hybrid methods combining offline training with online optimization are also sought after to enhance adaptability in real-time applications. Altogether, while several improvements are still expected, our current observations already yield several conclusions, paving the way to further investigations and holding promises for ML-based advanced processing techniques in incoherent optical systems.

First, we have noticed that despite long-term experimental instability hampering the forecast of spectral correlation maps from the seed parameters, the average DFT photodiode response (i.e. average spectrum) enabled excellent prediction of spectral fluctuations (see Fig. 7). Previous works demonstrated the capability of ANNs for successful data analysis in incoherent systems such as predicting temporal intensity features from spectral intensity only[48,50] (i.e. without phase information). In the current study, we demonstrate that the statistical properties of incoherent signals (i.e. spectral fluctuations) can be forecast by an ANN trained from averaged measurements only. The underlying mechanism involved in such a macroscopic approach leveraging averaged data is not fully determined yet. One may thus wonder whether these results can be generalized in a broader context of MI dynamics and nonlinear instabilities. However, ANNs trained from both numerical and experimental data demonstrated their potential for accessing incoherent signal properties without the need for implementing complex and costly real-time measurement architectures. Similarly, from a numerical viewpoint, these results should alleviate the requirement for time-consuming Monte-Carlo simulations in the study of incoherent nonlinear processes.

Secondly, the numerical results reported from two-seed scenarios demonstrate minimal errors, highlighting the prediction and inference accuracy of our ANN method. Yet, an increased error in the four-seed scenario (both numerically and experimentally) indicates that a rise in the complexity and dimensionality of the system reduces the network inference accuracy of the seed parameters, but not necessarily the prediction accuracy of the output waveform fluctuations (at least numerically). Experimentally, considering the resolution of the wave-shaper and assuming perfect environmental stability, there is over $10^{11}$ possible combinations for two-seed optical control. This value rises to over $10^{22}$ when considering four optical seeds, an important figure compared to the mere $<10^5$ cases tested for ANN training.

In this view, the limited inference quality of the input conditions for increased dimensionality questions the unicity of the nonlinear transfer function observed for such incoherent MI processes. More importantly, it interrogates the ability of the current detection scheme (DFT resolution, dynamic range) and selected observables (average spectral intensity and corresponding Pearson correlations) to lift any ambiguity between different seeding conditions. The proof-of-concept presented here however shows potential for concealing data through nonlinear noise amplification and chaotic dynamics—an approach similar to incoherent cloaking. In particular, the network's ability to access and retrieve hidden information, such as the phase of weak signals in a noisy environment, which is not discernible through a standard approach or spectral measurements, presents a promising area for further investigation.

On a third note, our results demonstrate the potential to control and reshape spectral fluctuations properties within a broadband coherent signal generated nonlinearly from noisy input signals. The approach thus shows that the statistical nature of these spectral fluctuations can be tuned qualitatively and quantitatively via all-optical seeding. We thus show that a neural network can be leveraged to predict complex and coherent transfer functions. Importantly however, complementary numerical studies illustrate that such ANN-

fuelled approaches can be used to establish and select suitable initial conditions for generating tailored incoherent signals, with several orders of magnitude speed-up compared to GNLSE-based simulations (see Supplementary Information, Fig. S5).

In line with these observations, it is worth noting that knowledge distillation and generalization in ANNs, are particularly promising when aligned with established MI analytical models[53–55,65,79–82]: propagation dynamics can be normalized through key parameters of the GNLSE such as dispersive and nonlinear lengths[54,74], so that MI seeding and associated evolution may be fully parametrized via normalized frequency detuning (i.e. via normalized MI, gain cutoff), phase, and modulation depth. Training ANNs on these normalized representations should allow distilled models to retain essential MI features and transfer key physical insights to smaller and more efficient models[83]. In this framework, normalization should thus enable generalization across different propagation regimes—such as varying propagation lengths, pump powers, pulse durations, and seed strengths—by decoupling dynamics from specific system configurations.

In this study, training data were limited to fixed pulse powers and durations for both pump and seeds (propagating in a limited fibre length). Expanding this initial parameter space (by e.g. varying pump and seed characteristics) could enable ANNs to learn physics-informed features that generalize across different MI propagation scenarios, even under evolving noise conditions. For instance, such ML approaches could potentially be adapted to more advanced analytical frameworks, extracting physical quantities such as eigenvalues within the inverse scattering transform (IST)[55,79–81] - albeit within the theoretical constraints of a pure NLSE. Ultimately, such hybrid approaches could improve model robustness and reliability for a broader range of MI regimes, but might also bring about powerful tools for analyzing and classifying noise in MI, such as predicting highly localized structure formations or tailoring specific MI evolution for a variety of real-world applications.

Finally, we note that in the current experimental proof of principle, we have focused our study to the reshaping of the first MI sideband fluctuations (due to the sensitivity limitations of our photodiode within the DFT scheme). However, improved measurement schemes previously demonstrated the impact of optical seeding in MI cascaded processes[57]. We therefore expect that ANN frameworks can readily help in the reshaping of incoherent FWM conversion for advanced signal processing. In particular, we anticipate that such an approach could be leveraged for spectrally mixing and interleaving complex cascaded processes for the generation of optical signals with tailored statistical properties. We also expect that examining multidimensional propagation scenarios featuring higher complexity (e.g. leveraging multimode fibres and intermodal nonlinear interactions[17,40,84]) could also benefit from such deep learning approaches for gaining fundamental insight into higher-dimensionality incoherent dynamics while unlocking new potential for on-demand and reconfigurable photonic signal generation.

## Methods
### Experimental setup
Optical pulse preparation starts from an initial femtosecond pulse train emitted at 50 MHz repetition rate from a commercial laser source (Menlo Systems, C-fiber HP Custom). The laser generates a pulse at a central wavelength of 1550 nm, with a duration of 80 fs (FWHM), an average power of 82 mW, and features a spectral coverage from 1480 to 1640 nm (at −20 dB cut-off). The initial broadband laser pulse is thus filtered and reshaped into picosecond pulses via a programmable spectral filter (Finisar–Waveshaper, 4000A, C band).

The Waveshaper is leveraged to generate a ~12 ps pump pulse centred at 1556.8 nm (by applying a 74 GHz bandpass filter). The Waveshaper is also used to generate and tailor (coherent) optical seeds originating from the same initial pulse. A number of seeds (2 or 4) are

filtered using the same process, with a continuously adjustable wavelength within the range 1551.3–1556.3 nm. In this case, the seeds are twice longer than the pump (of ~24 ps obtained from a 37 GHz bandpass filter) to ensure convenient temporal overlap between all filtered signals. The seeds also feature an additional attenuation of −25 dB compared to the pump and an arbitrary relative spectral phase (ranging from 0 to 2π) can be added to the seed signals. After filtering, the pump power is ~50 μW and each respective seed is ~3.5 μW.

These signals are then amplified by an Erbium-Doper Fibre Amplifier (Keopsys, EDFA–PEFA-SP) with a fixed current to reach an average output power of ~10 mW and ensure suitable nonlinear pulse propagation in the fibre. The Amplified Spontaneous Emission (ASE) of the EDFA is also filtered within the range 1544.5–1557.5 nm using a bandpass filter (CDWM SM-1551–AFW Technologies) and used as a source of white noise nonlinearly amplified together with the coherent seeds during subsequent fibre propagation. Once prepared, the amplified signal is injected into a Highly NonLinear Fibre (HNLF) of 385 m with dispersion parameters $\beta_2 = -1.78$ ps$^2$ km$^{-1}$ and $\beta_3 = 0.07$ ps$^3$ km$^{-1}$ measured at a central wavelength of 1550 nm. The nonlinear coefficient is 8.4 W$^{-1}$ km$^{-1}$ and the overall losses are 6.1 dB during propagation (including splices and linear attenuation).

After nonlinear propagation and spectral broadening, the output waveform is sent to a 90:10 fibre beam splitter. 10% is sent to an Optical Spectrum Analyzer (Anritsu, OSA–MS9710B) for monitoring the average spectrum with a spectral resolution of 0.07 nm. The remaining 90% is sent in a second programmable spectral filter (Finisar–Waveshaper, 4000 A, C + L band) to filter a region of interest within the spectral range [1557–1583 nm] and attenuate the rest of the spectrum. Afterwards, the filtered output signal is injected into a Dispersion Compensating Fibre (DCF) with a dispersive factor of $D = 420$ ps/nm and 3.8 dB insertion loss at 1550 nm.

After DCF propagation, the temporally-stretched spectrum can be readily measured in real-time through ultrafast detection systems ensuring adequate temporal resolution for the time-frequency mapping associated with the DFT measurement technique[47,57]. Here, the temporally-stretched spectrum is measured with a 5 GHz InGaAs biased photodiode (Thorlabs, DET08CFC/M) connected to a 6 GHz Real-Time Oscilloscope (RTO–Rhode & Schwarz, RTO2064). The RTO is synchronized with an external clock reference at 10 MHz (Stanford Research Systems–FS752) also driving the input laser oscillator used as an RTO signal trigger. Together, this detection scheme allows measuring the DFT spectra with 200 ps resolution (at 20 GSa/s) and minimal jitter, which yields an equivalent spectral resolution of 0.48 nm for real-time DFT measurements (and an equivalent spectral sampling rate of 0.12 nm per point). The overall experimental process, for both updating the Waveshaper seeding parameters and for the acquisition of 1000 DFT traces, takes approximately 3.5 seconds, so that our experimental framework can acquire ~1000 seeding scenarios per hour of measurement.

We note that the filtering stage before the DCF guarantees the suppression of the pump (and the lower wavelength region where optical seeding is implemented) which in fact ensures (i) a reduced power and thus a pure linear propagation in the DCF, (ii) an absence of temporal overlap between spectral components from adjacent pulses after DFT, i.e. 20 ns time period (iii) an absence of photodiode saturation (average power of 50 μW at the DCF output) along with an optimal use of the RTO dynamic range, and (iv) limited measurement artefacts due to the photodiode impulse response.

### Numerical simulations
Our numerical study relies on the generation of a substantial dataset obtained by solving the Generalized Nonlinear Schrodinger Equation (GNLSE) for the propagation of an optical pulse through a Highly Nonlinear Fibre (HNLF). We use realistic simulation parameters considering the HNLF used in experiments, with dispersion parameters

$\beta_2 = -1.78$ ps$^2$/km, and $\beta_3 = 0.077$ ps$^3$/km, and a nonlinear parameter $\gamma_O = 8.4$ W$^{-1}$ km$^{-1}$. In our simulation we initially generate a Gaussian pump pulse derived from filtering a 80 fs laser pulse with a central wavelength $\lambda_O = 1560$ nm and a spectral bandwidth of 15 GHz (FWHM). Using the same approach, we also generate a set of 2 (or 4) seeds, each assigned a random central wavelength in the range [1562.5–1564.6] nm, and a random spectral phase in the range [0 – 2π]. The spectral bandwidth of the seeds is also 15 GHz, but the seeds are attenuated by −50 dB compared to the pump.

This constitutes our initial seeded pulse, to which we add a broadband input noise competing with the coherent seeds during nonlinear MI amplification. White noise is included through the addition of 10 photons per spectral bin[74] (with random spectral phase) and ASE noise is included with a higher level of 100 photons per spectral bin over the range 1559.4–1564.6 nm, covering the spectral region where the seeds are implemented. This signal is amplified to reach a peak power of 1.77 W and propagated over 485 m of the above-mentioned HNLF by solving the GNLSE using the split-step method[74].

Monte-Carlo simulations are performed by repeating the propagation for 500 different realizations considering the same seeding parameters but different input noise. These different realizations are used to reconstruct the spectral correlation map using a Pearson metric. Such 500 Monte-Carlo realizations for each seeding scenario are typically performed in ~12 s (i.e. ~260–320 ms/simulation/core) for CPU-based parallel computing of the GNLSE with up to 16 cores. Repeating this process for an ensemble of different seeding conditions (i.e. wavelength and phase), we can generate two extended numerical datasets: one dataset for 2 seeds, comprising 90,000 random seeding conditions, and another dataset for 4 seeds, comprising 105,000 random seeding cases.

### Numerical data standardization

The neural networks operate with predictions ranging from 0 to 1, data must therefore be preprocessed and standardized specifically to this end:

**Correlation map preprocessing.** The 2D correlation map matrices (obtained from 500 Monte-Carlo simulations) are reshaped into 1D vectors suited for 1D layers FFNN. The values of Pearson correlation are linearly rescaled to [0, +1] values instead of [−1, +1]. Correlation maps obtained from GNLSE simulations are down-sampled from $1024 \times 1024$ pixels to $128 \times 128$ pixels over the selected region of interest, i.e. 1540–1580 nm, corresponding to a spectral resolution of 0.31 nm.

**Average spectrum preprocessing.** The average spectrum from 500 realizations is also normalized between 0 and 1 via a direct mapping of the logarithmic scale spectrum (1024 points within a 40 nm spectral span). In this case, we employ a 50 dB dynamic range spectrum to preserve the network's ability to forecast the general dynamics in the second and third sidebands of the MI spectrum (knowing there is a typical 10 dB signal drop between two adjacent sidebands) while ensuring that the central part of the spectrum (i.e. the pump experiencing SPM) does not get overly rewarded by the network during the training phase.

**Seed parameters preprocessing.** Each input seed possesses two parameters: its wavelength (in nm) and its spectral phase (between 0 and 2π). For a given seeding configuration, the two (or four) seeds are therefore ranked according to their wavelength value ($\lambda_1, \lambda_2,...$) and this wavelength $\lambda_i$ is normalized (between 0 and 1) with respect to the spectral interval where they are randomly distributed within the dataset (i.e. the range 1562.5–1564.6 nm). The spectral phase parameters are encoded in terms of phase difference between the previously ranked seeds (e.g. $\Delta\varphi_{2,1} = \varphi_2 - \varphi_1$) to limit phase wrapping issues during ANN training. For each phase difference in the network, we also use a sine and cosine projection and normalize their values (between 0 and 1) so that, after training, the phase can be readily retrieved with a *tan* function.

### Experimental data standardization

The dataset extracted from experiments is preprocessed in a similar manner as numerical data. Average spectra and corresponding spectral correlation maps are directly extracted from 1000 experimental measurements using sequential DFT acquisition. Both the spectrum and the maps are obtained over a specific 10 nm region of interest corresponding to the filtered DFT spectral range: The spectra and correlation maps are extracted between 1556 and 1566 nm, with a corresponding spectral sampling rate of 0.12 nm/point. The spectra possess a length of 82 points and are normalized via linear scale to account for the limited dynamic range of DFT measurements. Experimental correlation maps feature $82 \times 82$ pixels and are normalized in the same way as the numerical dataset. The seed parameters, experimentally encoded on the Waveshaper, are also processed in the same manner as in numerical simulations. However, in our experiments, the seeds are set within the range 1551.3–1556.3 nm (i.e. on the short wavelength side of the pump).

### Neural network architecture and training

The implemented Artificial Neural Network (ANN) is a Feed-Forward Neural Network (FFNN) encoded in Matlab via its Deep Learning toolbox. The solver used for training is a Stochastic Gradient Descent with Momentum (SGDM) with an initial learning rate between 0.015–0.018, and up to 700 epochs depending on the dataset feeding the network (see Table S5 in the Supplementary Information for further details). For all ANN inferences and predictions reported in the manuscript, the network is a FFNN composed of a total of 5 layers (1 input, 1 output and 3 inner layers). Depending on the nature of the data forecast and used for training (i.e. length of the input/output vectors), each FFNN layer width is adjusted and the overall number of neurons (perceptrons) thus ranges from 1676 to 37,392 (see Table S4 in the Supplementary Information for details on each FFNN architecture).

In order to assess the network's successful training (e.g. without overfitting), we evaluate the prediction error of the trained network within a validation dataset for which the network was not trained. This validation subset is randomly extracted using 5% of the experimental dataset and 10% of numerical dataset, respectively, and the Root Mean Square Error (RMSE) provided in the manuscript corresponds to the values computed on this validation subset (i.e. between the ANN predictions and the ground truth extracted from numerical or experimental data). For instance, the RMSE of the correlation maps is obtained by averaging each pixel RMSE over the whole map, while the spectrum RMSE also corresponds to the mean RMSE over the spectral trace (after normalization). The global error of the network training is computed as the RMSE average ensemble of all the cases predicted within the validation subset (i.e. untested 5–10% of the global dataset).

## Data availability

The numerical and experimental data used for artificial neural network training that support the findings of this study are available in Zenodo[85] with the identifier https://doi.org/10.5281/zenodo.15179897. The figure data generated in this study are provided in the Source Data file. Source data are provided with this paper.

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

## Acknowledgements

This work received funding from the European Research Council (ERC) under the European Union's Horizon 2020 research and innovation programme under grant agreement No. 950618 (STREAMLINE project - B.W.) and No. 947603 (QFreC project - M.K.). This research benefited from the support of the Platinom platform, with funding from the European Union and the Nouvelle Aquitaine council under the PILIM program. B.W. acknowledges the support of the French ANR through the OPTIMAL project (ANR-20-CE30-0004), and the Nouvelle Aquitaine council (SPINAL project). We furthermore acknowledge support from the German Research Foundation (Deutsche Forschungsgemeinschaft; DFG) within the cluster of excellence PhoenixD (EXC 2122, Project ID 390833453). The authors thank R. Dauliat and P. Roy for HNLF fibre drawing as well as V. Kermène and V. Couderc for their support and fruitful discussions.

## Author contributions

Y.B. led the project and carried out the experiment with main contribution from L.S. and V.T.H.; B.P.C., A.B. and M.F. further contributed to the implementation and validation of the setup; Y.B. analyzed the data with main contributions from L.S.; M.K. and B.W. developed the original idea and initiated the work; A.T. and J.M.D. contributed to scientific discussions; BW supervised the project. All authors contributed to the manuscript preparation.

## Competing interests

The authors declare no competing interests.
