## [Transparent Peer Review file · Nature Communications]

Deep learning prediction of noise-driven nonlinear instabilities in fibre optics

Corresponding Author: Dr Benjamin Wetzel

Version 0:

Reviewer comments:

Reviewer #1

(Remarks to the Author)

The paper deals with the application of deep learning techniques for prediction of noise-driven nonlinear instabilities in fibre optics. It is demonstrated that coherent optical seeding can be used to tailor incoherent spectral broadening. The main result related to deep learning application is the demonstration that the artificial neural networks are capable to learn and predict relevant complex incoherent dynamics, both numerically and experimentally. Other interesting results include analysis of the possibility to infer input optical seeds properties from incoherent signals measured at the fibre output and forecasting the fluctuation characteristics of the nonlinearly-broadened output spectrum.

I think this is an interesting and solid work that deserves to be published in Nature Communications. I would suggest to address the following points in the revision:

1) As often in application of machine learning (ML) methods for particular applications, it would be useful to make more clear for non-specialists in the field (say, in bullet style listing, or table) what is either not possible without using ML or what are quantitative advantages of using ML compared to numerical modeling or other methods of data/signal processing. More clear and focused listing would be useful.

2) It is not fully clear in terms of comparison of the efficiency of different methods how much time the learning process takes. Considering scenario with changed parameters, or changing environment, how much knowledge distillation can take compared to non-ML methods?

3) Some discussion around the choice of a feed-forward neural network might be useful to guide readers. There are numerous types of artificial neural networks tailored to different tasks. Is FFNN the best/well suited for this task? For applications of ANNs in changing environment (that might be of specific interest for the applications in experiments) deep reinforcement learning has been used recently in other types of photonic applications. Would this be a good direction for the considered application? I would suggest to extend part related to the perspectives that this work offers to the field.

Overall, this is high quality work on the interesting and expanding topic. I recommend to publish this paper.

Reviewer #2

(Remarks to the Author)

REVIEW NATURE COMMUNICATIONS

The work claims to demonstrate that coherent optical seeding can be used to tailor incoherent spectral broadening by exploiting deep learning methods capabilities. In specific, the work is centred in the regime in which coherent optical seeding and broadband noise are competing during modulation instability when propagating through a nonlinear fibre to study and assess the artificial neural networks (ANN) – based approach in three different levels.

In the first place, to infer the optical seeding from the incoherent output of the fibre (wavelength and phase) despite the weak nonlinear signal amplification. Secondly, to forecast the fluctuation characteristics of the nonlinearly-broadened output spectrum. Finally, to study the aptitude of the ANN to optimize specific correlation features, tailoring incoherent waveforms, by optical seeding and deep learning. In order to achieve all this, the paper describes and explains in a rigorous manner the followed methodology to demonstrate the usefulness of ANN for nonlinear instabilities study and setting the path to tailored nonlinear spectral fluctuations.

Major comments

The paper is well explained and logically ordered to follow and understand the results and conclusions of the study. A suggestion to improve the quality of the work would be to explain more in detail which are the differences in training when presenting the results of inferring the seed inputs or forecasting the outputs using ANN. It is not clear in the results section which is the difference in training for both studies, as lines 110-112 in the main text talk about forward or backward propagation training and in subsection "Forecast of incoherent nonlinear dynamics from artificial neural networks" the training is described but in subsection "ANN prediction of incoherent spectral fluctuations" is not. It would be desirable to provide the same amount of information for all the analysed cases presented in the results section. In specific, regarding the results of the ANN when retrieving the phase of the experimental 4-seed scenario in the subsection "Experimental validation of incoherent nonlinear dynamics inference via ANNs", it could be explained the existent 24.3% of error when retrieving the seed phase (provided in Table S2 in Supplementary Information) and explain why is so high in comparison to 2-seed scenario and compared to numerical data. Related to this, for coherence of the work to the numerical counterpart, the phase map $\Delta\phi_{(3-4)}$ should be displayed in Figure 4 and explained in the text together with the previous suggestion.

Minor comments

Some errors were found in the main text:

Line 183, in the description of Figure 3, it seems to lack some part of the sentence as well as the reference to panel b.

Line 199, it is stated "red line" when the figure shows "blue line".

Line 368, in main text it is stated 8.1% meanwhile in Table S2 in Supplementary Information it is stated 8.7%.

Line 400, referenced Fig.5 instead Fig.6.

Conclusion

The paper shows the capability, as proof of concept, of using an ANN to infer and forecast nonlinear dynamics both from numerical simulations, agreeing excellently with the simulation results from the Monte Carlo method, as well as, from experimental data, regarding the challenge of needing large experimental datasets and dealing with short-term stability of the experimental set-up and measuring in a regime of weak power of seeds in presence of noise.

The paper provides a rigorous methodology to be reproduced, with exception of the previous suggestions. The work shows an improvement from the state of the art introducing the use of ANN to help in the accessing of incoherent signal properties, as well as, to demonstrate the potential to control and reshape spectral fluctuations properties within a broadband coherent signal generated nonlinearly from noisy input signals, and thus, speeding up the tailoring of incoherent signals. Furthermore, the paper proposes a method to enhance optimization time and data generation costs using this methodology.

Reviewer #3

(Remarks to the Author)

Version 1:

Reviewer comments:

Reviewer #1

(Remarks to the Author)

The authors made a comprehensive revision of the manuscript that substantially improved the clarity of the presentation.

Changes are appropriate.

Overall, this is an engaging and well-written paper that contributes new knowledge to the field. I recommend this manuscript for publication.

Detailed reply to the referees

We thank both reviewers for the very positive assessment of our manuscript. Both referees highlight the high quality, rigor, interest, and clarity of our manuscript, and thus recommend its publication in Nature Communications after addressing their points.

We provide below a detailed response to both referees' comments and suggestions. We uploaded a revised version of our manuscript, where the redlined text highlights the modifications made to address the reviewers' comments and improve the manuscript's clarity. We further amended our Supplementary Material file to address the points raised by the reviewers in detail. In parallel, we also updated the manuscript to fully comply with the editorial policies of Nature Communications.

Referee: 1

The paper deals with the application of deep learning techniques for prediction of noise-driven nonlinear instabilities in fibre optics. It is demonstrated that coherent optical seeding can be used to tailor incoherent spectral broadening. The main result related to deep learning application is the demonstration that the artificial neural networks are capable to learn and predict relevant complex incoherent dynamics, both numerically and experimentally. Other interesting results include analysis of the possibility to infer input optical seeds properties from incoherent signals measured at the fibre output and forecasting the fluctuation characteristics of the nonlinearly-broadened output spectrum. I think this is an interesting and solid work that deserves to be published in Nature Communications. I would suggest to address the following points in the revision

Recommendation: Overall, this is high quality work on the interesting and expanding topic. I recommend to publish this paper.

Reply: We thank the referee for their kind review, underlining the quality and interest of our manuscript and the results reported. The referee recommends publication of the manuscript after addressing a few points in the revision. We address below the comments of the referee.

Comment A1: As often in application of machine learning (ML) methods for particular applications, it would be useful to make more clear for non-specialists in the field (say, in bullet style listing, or table) what is either not possible without using ML or what are quantitative advantages of using ML compared to numerical modeling or other methods of data/signal processing. More clear and focused listing would be useful.

Reply to Comment A1:

We thank the reviewer for this constructive suggestion. The referee raises an excellent point that is of course difficult to quantify definitively, as research is advancing very fast in this topic and new approaches are regularly being developed and announced. However, we fully agree that a general summary for non-specialists of what machine learning (ML) offers would strengthen the manuscript, and as suggested, we have now added a short discussion in the revised version while including a longer table in the Supplementary Material (Table S1) summarizing the advantages and limitations of ML approaches in comparison to traditional numerical and signal processing methods, in the framework of nonlinear fibre optics and, in particular for studying and controlling modulation instability processes.

Along this line, we have also added several references to complete the previous overview of the state of the art on this topic and further refer the interested reader to several recent review article dealing with the topic.

While ML holds great promise, especially in complex experimental regimes, we also note in the revised text that its ultimate performance remains dependent on data quality, experimental reproducibility, and the specific task at hand. For this reason, we have opted not to make strong claims but instead aim to highlight the complementary role of ML in advancing the study and control of MI.

Comment A2: It is not fully clear in terms of comparison of the efficiency of different methods how much time the learning process takes. Considering scenario with changed parameters, or changing environment, how much knowledge distillation can take compared to non-ML methods?

Reply to Comment A2: The comment of the referee is indeed very interesting. We agree that the training time associated to ANN learning process was not clearly addressed in the initial version of the manuscript. We have now added a description of the ANN architecture and training time for the different ANN scenarios shown in the manuscript. Specifically, we added a section with two Tables in the Supplementary Information (Table S4 and S5), respectively summarizing the ANN network architectures and associated training features/times for each of the figures of the main manuscript. In Table S4, one may observe that the ANN training can take between 40 minutes up to almost 10 hours, depending on the overall number of perceptrons in the network (up to $> 37,000$) and the number of epochs considered (i.e. the number of times training see the same whole $> 50,000$ dataset – up to 700 epochs in our case). Of course, depending on the accuracy required for a specific application, as well as the intrinsic data quality and size, these training (hyper)parameters could be adjusted to speed up the process. Typically, for the various scenarios considered, we found empirically that reasonable accuracy can be obtained for ANN training below 2 hours. In such cases, there is of course room for improvements and further training optimization. However, this should be put into perspective with the duration of large datasets generation from numerical simulations (i.e. $\sim 100,000$ numerical seeding scenarios of 500 Monte-Carlo simulations each) or their acquisition from experimental measurements (i.e. $\sim 60,000$ experimental seeding case with 1000 DFT measurements each), typically taking ~ 60 hours (for experiments) and up to two weeks (for GNLSE simulations). For completeness, we now have also specified experimental and simulations time for data acquisition in the manuscript.

In this view, knowledge distillation and generalization in ANN training might be of particular interest, especially in view of established analytical MI models. Modulation instability can be normalized using the generalized nonlinear Schrödinger equation (GNLSE), incorporating key parameters such as the dispersive length (L_D) and nonlinear lengths (L_{NL}), as well as MI modulation parameters such as normalized frequency modulation detuning (a) with respect to MI gain cutoff (Ω_c), and seed relative strength/modulation depth (a_{mod}). By leveraging knowledge distillation, a trained ANN should potentially transfer insights gained from these normalized representations, ensuring that the distilled model retains essential MI characteristics that can then be transferred to train a smaller model based on these normalized features.

A significant advantage of such a normalization is its potential for generalization across different parameter regimes: e.g. different propagation length or input pump properties (power, duration) or seed relative power for a particular fibered configuration. With appropriate normalization, relevant neural networks should then generalize trainings to different seed relative intensities, propagation lengths, and nonlinearities in a straightforward manner. However, in the framework of this study, the dataset generated was performed at a given pump power and duration, within a single fiber of fixed length, and with a fixed seed relative power with varying detuning and spectral phase. In this view,

only two “normalized parameters” in the physical sense of MI (i.e. spectral seed detuning and relative phase) may typically be extracted, making knowledge distillation fairly limited for this specific study. This choice of fixing the parameters was done to ensure optimal experimental stability, stable noise features, and well-defined power ratio between incoherent noise and coherent seeds, both feeding MI amplification. It is however clear that extending our strategy of coherent optical seeding to a larger parameter space (varying both pump and seed power, duration, etc.) should allow the model to learn underlying physics-informed representations that are less dependent on specific parameter values and more transferable across different MI scenarios, especially for changing input noise characteristics. Yet, one of the main challenges in practical implementation consists in assessing the impact of experimental noise on MI processes. The stability and robustness of neural network predictions under different noise properties so far remain difficult to quantify systematically.

Looking forward in the future, ML with the help of knowledge distillation and physics-informed neural networks might be a potential way to analyze and classify the noise properties itself, or extract particular features in noise-driven MI leading to the development of specific propagation scenario (e.g. taming or triggering highly localized structures and extreme event formation). For instance, it is possible that such ML techniques can be implemented for e.g. eigenvalues extraction within the framework of Inverse Scattering Transform (IST) theory (while theoretically constrained to pure NLSE). This could lead to a novel approach where ML-driven noise characterization may enhance the reliability of MI modeling for sensitive applications. This combined approach may also improve robustness against experimental variations and facilitate reliable predictions across diverse MI conditions.

In summary, we fully acknowledge the potential of these advanced training strategies, which will be subject to future investigations, but falls outside the scope of the current study given the generated dataset. In line with the referee’s comment, we have however added a discussion on the potential of knowledge distillation and physics-informed networks for the generalization of such ML approaches with respect to MI theory and normalized modulation parameters.

Comment A3: Some discussion around the choice of a feed-forward neural network might be useful to guide readers. There are numerous types of artificial neural networks tailored to different tasks. Is FFNN the best/well suited for this task? For applications of ANNs in changing environment (that might be of specific interest for the applications in experiments) deep reinforcement learning has been used recently in other types of photonic applications. Would this be a good direction for the considered application? I would suggest to extend part related to the perspectives that this work offers to the field.

Reply to Comment A3:

We agree with the referee. Our method, involving feedforward neural network (FFNN) aimed to serve as a proof of principle for a simple and standardized architecture. Specifically, we employed a fully connected network with three hidden layers, varying only in the dimensionality of input, inner, and output layers to adapt to different scenarios. The main goal was to establish a reproducible and interpretable framework using MATLAB, thus ensuring accessibility through a well-documented toolbox.

In this study, we found that the accuracy of ANN-based training and associated prediction is largely dictated by the quality of input data. Yet, there are clear opportunities for improvements in terms of network architecture and hyperparameter tuning, depending on whether the focus is on fundamental studies (e.g. physics informed networks) or applied aspects relying on e.g. online optimization (heuristic search, reinforcement learning strategies, etc.).

As fully-connected FFNNs were previously demonstrated in a variety of experiments in nonlinear fiber optics (see Genty, G. et al. Machine learning and applications in ultrafast photonics. Nat. Photonics 15, 91–101, 2021), we used this ANN architecture as a robust benchmark in this study to guarantee a certain degree of universality in the network trained. However, we have also explored various ML techniques and ANN architectures tailored to different types of input-output data. While out of the scope of this study, initial works based on autoencoder have shown promises for training models based on initial seed conditions, while Convolutional Neural Networks (CNNs) were succinctly tested for processing and training on correlation maps (considering these maps for ML strategies based on image processing tools and pattern recognition techniques), with a relative success that could be further explored in subsequent works. In particular, transitioning to Python-based implementations leveraging specialized and more flexible machine learning toolboxes seems particularly interesting for future in-depth studies focusing on the specific network architectures and conformations. Once refined, these strategies may likely provide better accuracy or training speed depending on the datatype considered. More advanced architectures such as Generative Adversarial Networks (GANs) also hold further promise for optimizing both prediction and inference models in a unified framework.

In this context, we now provide our numerical and experimental data used for ANN training in full, within an open data repository (Zenodo – [doi:10.5281/zenodo.15179897](https://doi.org/10.5281/zenodo.15179897)), so that that readers have access to the data and have the opportunity to explore the specifics and efficiency of different ANN architectures, a focus that goes beyond our manuscript's work.

When considering experimental conditions, especially for optimizing noisy and dynamically evolving experimental conditions, Reinforcement Learning (RL) constitutes a powerful tool for online optimization of complex, nonlinear and chaotic systems. For instance, RL strategies were recently demonstrated numerically for suppressing modulation instability in spatial beams (see Kalmykov, N. I. et al., Suppressing modulation instability with reinforcement learning. Chaos Solitons Fractals 186, 115197, 2024). Such a promising approach provides a clear interest for advanced MI control, but would however requires – in this fiber optics case – controlled modulation of the pulse throughout fiber propagation, involving e.g. non-uniform propagation medium (longitudinally-adjusted fiber parameters (Mussot A. et al. Advances in Optics and Photonics, 10, 1, 2018) or modulation in a fiber recirculating loop (Copie F. et al. Optics Letters 47, 14, 2022). While this particular technique may be difficult to implement for reconfigurable and online optimization, we believe that reinforced learning constitutes an excellent strategy for mitigating changes in environmentally-unstable conditions. In our case, this would for instance be an ideal way to consider and mitigate both thermal drifts and EDFA instabilities in realistic experimental conditions, especially for demanding applications.

In line with the previous comment of the reviewer, hybrid methods combining offline training with online optimization is also sought after to enhance adaptability in real-time applications. Finally, as discussed earlier, physics-informed learning and data-driven approaches may also provide interesting perspectives to gain insight in the dynamics and thus improve knowledge distillation and interpretability of ML predictions and inferences.

Along these comments, we have now added a brief discussion on the choice of feed-forward neural network architectures to guide readers and further extended the Discussion section related to the perspectives of this work, especially in view of other ML strategies applicable to MI control and optimization.

Referee: 2

The work claims to demonstrate that coherent optical seeding can be used to tailor incoherent spectral broadening by exploiting deep learning methods capabilities. In specific, the work is centred in the regime in which coherent optical seeding and broadband noise are competing during modulation instability when propagating through a nonlinear fibre to study and assess the artificial neural networks (ANN) – based approach in three different levels.

In the first place, to infer the optical seeding from the incoherent output of the fibre (wavelength and phase) despite the weak nonlinear signal amplification. Secondly, to forecast the fluctuation characteristics of the nonlinearly-broadened output spectrum. Finally, to study the aptitude of the ANN to optimize specific correlation features, tailoring incoherent waveforms, by optical seeding and deep learning. In order to achieve all this, the paper describes and explains in a rigorous manner the followed methodology to demonstrate the usefulness of ANN for nonlinear instabilities study and setting the path to tailored nonlinear spectral fluctuations.

The paper is well explained and logically ordered to follow and understand the results and conclusions of the study.

The paper shows the capability, as proof of concept, of using an ANN to infer and forecast nonlinear dynamics both from numerical simulations, agreeing excellent with the simulation results from the Monte Carlo method, as well as, from experimental data, regarding the challenge of needing large experimental datasets and dealing with short-term stability of the experimental set-up and measuring in a regime of weak power of seeds in presence of noise.

The paper provides a rigorous methodology to be reproduced, with exception of the previous suggestions (*see referee comments below*). The work shows an improvement from the state of the art introducing the use of ANN to help in the accessing of incoherent signal properties, as well as, to demonstrate the potential to control and reshape spectral fluctuations properties within a broadband coherent signal generated nonlinearly from noisy input signals, and thus, speeding up the tailoring of incoherent signals. Furthermore, the paper proposes a method to enhance optimization time and data generation costs using this methodology.

Reply: We are grateful for the referee's time in assessing our manuscript and for their positive comments. Below, we address in detail the points raised by the reviewer.

Comment B1: A suggestion to improve the quality of the work would be to explain more in detail which are the differences in training when presenting the results of inferencing the seed inputs or forecasting the outputs using ANN. It is not clear in the results section which is the difference in training for both studies, as lines 110-112 in the main text talk about forward or backward propagation training and in subsection "Forecast of incoherent nonlinear dynamics from artificial neural networks" the training is described but in subsection "ANN prediction of incoherent spectral fluctuations" is not.

Reply to Comment B1:

We fully agree with the referee's comment and thank the referee for this suggestion. The sentence of lines 110-112 indeed refers to "forward or backward propagation" and the wording may be unclear, so that the reader may understand that ANN respective approaches/architecture for inference and prediction of pulse propagation properties might be different. Here, we refer to "forward propagation" in the sense of nonlinear pulse evolution in the fiber, so that "backward propagation" implies using

the ANN to solve the “inverse problem” (i.e. using the ANN to simulate nonlinear backpropagation of incoherent pulse evolution from the output of the fiber to the input).

Solving the direct and inverse problem are typically referred to as prediction and inference in ML community, and this may lead to some confusion regarding the neural network strategy presented in our manuscript. In fact, the difference of ANN training for either inferencing the seed inputs or forecasting the outputs are marginal. The main difference arises from the type and vector size of the data respectively used in the input or output layer. In both cases however, the neural network used corresponds to a fully connected feed-forward network (FFNN) of constant depth, comprising 1 input layer, 3 (hidden) inner layers, and 1 output layer. The type of input vs output data for ANN training conditions the respective length of the input/output layer, and the key difference thus corresponds to adjustment of the respective widths of the 3 hidden layers (i.e. number of perceptrons per layer). These widths were adjusted for each case to ensure a smooth transition (i.e. expansion and the reduction of the layer widths) to match the input/output data size while ensuring an adequate number of perceptrons to train to avoid underfitting or overfitting.

For instance, when considering ANN training for a numerical dataset in a 2-seed scenario, the FFNN architecture is qualitatively similar for prediction and inference (5 layers, with progressively increasing/decreasing widths), as illustrated in the table below:

Training type	Input & output data used ANN training		ANN architecture: FFNN (number of perceptrons per layer)					Total ANN perceptrons
	Input	Output	Input	Inner #1	Inner #2	Inner #3	Output	
Prediction (forward propagation)	Seed parameters	Correlation maps	4	32	256	1024	16384	17700
Inference (backward propagation)	Correlation maps	Seed parameters	16384	1024	256	32	4	17700

Overall, the ANN architecture was selected to obtain a simple structure also providing a certain degree of reversibility between output prediction and input inference, by merely adjusting the data feeding the ANN. This was initially done to be able to assess and easily compare the ANN accuracy for an architecture to fit all the cases studied, thus reproducing nonlinear dynamics of different complexity, whether through a direct (prediction) or reversed (inference) transfer function provided by the ANN. We note that different or specific architectures could have been selected beyond standards FFNNs, with potentially better prediction or inference accuracy (see reply to Reviewer #1). In this paper, only slight empirical adjustments were made in the ANN training from one case to another, depending on the specific data used. For instance, marginal tuning of the learning rate (between 0.015 and 0.018) as well as the maximal number of epochs (between 100 and 700) was made to ensure training accuracy and stability while avoiding unnecessary computation time.

While these points were hinted in the initial manuscript, we agree that clearer explanations is needed to avoid any confusion for the reader. Along this line, we have expanded the Method section to explain these particular aspects while explicitly adding further clarifications in the main manuscript regarding the ANN training in the relevant sections. For completeness, we also added a section with two Tables in the Supplementary Information (Table S4 and S5) summarizing the ANN network architectures and training features for each of the studies reported in the manuscript.

Comment B2: It would be desirable to provide the same amount of information for all the analysed cases presented in the results section. In specific, regarding the results of the ANN when retrieving the

phase of the experimental 4-seed scenario in the subsection “Experimental validation of incoherent nonlinear dynamics inference via ANNs”, it could be explained the existent 24.3% of error when retrieving the seed phase (provided in Table S2 in Supplementary Information) and explain why is so high in comparison to 2-seed scenario and compared to numerical data.

Related to this, for coherence of the work to the numerical counterpart, the phase map $\Delta\varphi_{(3-4)}$ should be displayed in Figure 4 and explained in the text together with the previous suggestion.

Reply to Comment B2: We are grateful to the reviewer for their suggestion.

In our early neural network trainings, we indeed observed a relatively large RMSE (> 20% error) when trying to infer the input phase difference when considering both experimental data and a larger number of input seeds (i.e. 4 input seeds), as underlined in the supplementary material (Table S2, now Table S3). Along this line, we did not focus on these specific results in the initial version of the manuscript, and thus only worked on the inference of the seed wavelength in this particular scenario. We however agree that, for clarity, the same amount of information should be present between numerical and experimental data for an easier comparison. In this view, we have updated Figure 5 to include the inference of the phase information for 4 experimental seeds, now displayed in Figure 5(d). Along this line, we have also updated the number of inferred spectra displayed in Figure 5(c) to match those shown in Figure 3 (c).

We note that for consistency, we have therefore performed another ANN training to infer all the seed parameters in Figure 5(d), i.e. both wavelength and phase difference. The network used the same FFN layout, but we added 12 neurons in the output layer (yielding 16 instead of 4 neurons) corresponding to the inference of 6 combinations of phase differences between the 4 seeds, each projected in the complex plane (i.e. sine and cosine). In this case, the training results are consistent with those of the initial manuscript, where the predicted wavelengths exhibit a similar error (i.e. 15.4 % RMSE average for the 4 seeds) and the phase difference yield the same ~ 24.2 % RMSE average error over the 6 combination of seed phase difference.

While the training accuracy may appear relatively low in this experimental scenario, especially for the phase inference, it should be put into perspective with respect to both the use of numerical data (see e.g. Fig. 3) and whether considering a different number of seeds (see e.g. panels (b) and (d) in Figs. 3 and 5, respectively).

As summarized in Table S2 and S3 (in the revised SM version), and discussed in the main manuscript, the inference of 4 seed parameters is consistently less accurate than the inference of 2 seed parameters. Both experimentally and numerically, we observe that the inference error is always 2.5 up to 3 times higher when considering a larger input parameter for the seed inference (i.e. 4 seeds instead of 2). As previously mentioned in the initial version of the manuscript, we believe that this limitation (also seen from simulated data) question the unicity of the solution (i.e. whether a single correlation map can originate from a unique set of seed parameters) and the ability of the ANN to find such key input parameters from output spectral data generated with intrinsically limited resolution (and from dynamics highly sensitive to competing initial conditions).

Conversely, when comparing ANN seed inference obtained from either numerical and experimental data, we also observe a ~ 3 -fold degradation of the ANN accuracy when considering experimental dataset. As discussed in the manuscript, and further detailed in the Supplementary Material (see Figs. S3 and S4), we believe that this limited accuracy may be incurred to the experimental stability, especially for the seed parameter inference that is highly-sensitive to the weak EDFA power fluctuations.

Altogether, we believe that the results displayed in Fig. 5 now bring a consistent view on the quality and limitation of the ANN inference for a large input parameter space in realistic experimental conditions. The discussions above are consistent with the observed results, and also fully explain the relatively large error (24 % RMSE) observed in this scenario of phase seed inference. To clarify the

points discussed above for the reader, we also extended the discussion on the ANN inference accuracy in the main manuscript (from line 328 to 335) and updated the Supplementary Material.

Comment B3: Minor comments - Some errors were found in the main text:

- Line 183, in the description of Figure 3, it seems to lack some part of the sentence as well as the reference to panel b.
- Line 199, it is stated “red line” when the figure shows “blue line”.
- Line 368, in main text it is stated 8.1% meanwhile in Table S2 in Supplementary Information it is stated 8.7%.
- Line 400, referenced Fig.5 instead Fig.6.

Reply to Comment B3: We are extremely grateful to the reviewer for its careful and very thorough read of the manuscript. We agree that we overlooked a few typos, that have now been fully corrected (and highlighted in the redlined version of the revised manuscript).

Referee: 3

Reply: We are very grateful for the referee’s time in assessing our manuscript. We fully support this Nature Communications initiative to facilitate training in peer review and to provide appropriate recognition for ECR. We thank the referee for their comments and hope to have replied to all the points they may have raised in the other listed report(s).